# Sub- and Supra-Second Duration Perception of Implied Motion: Differences Between Athletes and Non-Athletes

**DOI:** 10.3390/bs14111092

**Published:** 2024-11-14

**Authors:** Weiqi Zheng

**Affiliations:** 1School of Psychology, Beijing Sport University, Beijing 100084, China; zhengweiqi@bsu.edu.cn; 2Laboratory of Sports Stress and Adaptation of General Administration of Sport, Beijing 100084, China

**Keywords:** implied motion, time perception, sub-second, supra-second, athlete

## Abstract

This study aimed to investigate the differences in duration perception between athletes and non-athletes when looking at implied motion images within sub- and supra-second time ranges. By adopting the temporal bisection method, the study analyzed the duration perception of 20 college student athletes and 20 non-athletes regarding the implied motion of daily life (running and walking) and static postures (standing). The results showed that the effect of movement posture was significant, i.e., the perceived duration of the implied motion posture was longer than that of the static posture. Specifically, athletes perceived longer durations in the supra-second time range compared to non-athletes, indicating that long-term training enhanced athletes’ time perception abilities. The findings provide new insights into the cognitive mechanisms of time perception and emphasize the influence of long-term physical training on temporal perceptual capabilities.

## 1. Introduction

Time perception is critical for everyday activities, encompassing both fundamental behaviors and refined movement coordination, such as in sports and exercise. In fact, whatever the discipline, the capacity for optimal sports performance is inherently contingent upon a superior ability of time perception, such as catching a badminton shuttlecock, shooting a basketball at the appropriate moment, and executing dance moves precisely [1]. Time perception reflects the duration and sequence of events and is essential for the conscious motor control of the pace and rhythm of actions.

Compared with non-athletes, elite athletes, regardless of their sport domains, exhibit better time estimation by showing higher accuracy and lower variability when reproducing the duration of scrambled pixels [2,3]. However, time perception may be influenced by the visual properties, potentially leading to a distortion of duration, as seen in both athletes and non-athletes. This study aims to examine how athletes perceive temporal durations in implied motion contexts compared to that of non-athletes.

Subjective time perception is often distorted by stimulus movement [4]. Instances of implied motion, such as a running animal or a swimming person, have been investigated as a special type of stimulus with processing mechanisms that are similar to those of real motion. Additionally, static images containing implied motion can also influence time perception [5,6]. For example, some studies have reported an elongation of perceived time for moving stimuli compared to static ones, as well as for stimuli with a higher degree of implied motion compared to those with less [7]. Recently, a growing body of research has focused on athletes’ duration perception of implied motion images depicting sports events. The implied motion of body gestures, regardless of their simplicity or complexity, necessitates precise timekeeping, which is crucial in competitive sports due to the importance of action anticipation, motor coordination, and motor synchronization for successful performance [8]. However, the results of these studies were not consistent. For instance, Liang, Zhang, and Zhang found that divers displayed higher accuracy in duration estimation for diving stimuli than novices did [9]. Jia, Zhang, and Feng suggested that sports experts perceive a longer duration when viewing expertise-related implied motion stimuli compared with others with no expertise/experience [10]. In contrast, Chen et al. reported that elite pole vaulters perceived time as shorter when observing highly dynamic images (such as a pole vault jump) compared to less dynamic ones (such as a fencing lunge) and scrambled pixels, indicating that action observation distorts individuals’ time perception by compressing the perceived passage of time [11].

Possible explanations for these inconsistent results include variations in time perception paradigms and the use of different implied motion images across various sports domains. Therefore, the current study aims to address this issue by adopting implied motion images depicting daily movements, such as running and walking, which differ from expertise-related stimuli. In addition, the study seeks to enhance our understanding of the effects of long-term exercise training on general cognitive abilities from a time perception perspective.

Moreover, this study investigated duration perception in both the sub-second and the supra-second time ranges. Previous research suggests that supra-second duration perception is more affected by cognitive processes than sub-second perception, which is primarily governed by automatic processes [12]. And supra-second tasks often engage attention and memory processes, which are thought to be more developed in athletes and may be enhanced through sports training. Based on this, we hypothesize that the duration perception of implied motion in the supra-second range will be more influenced by athletes’ long-term sports training than that in the sub-second range.

The temporal bisection task was chosen for its ability to provide robust duration perception measurements while minimizing the influence of explicit cognitive processing, which may introduce additional variability [13]. Although a reproduction task might reveal cognitive distortions in athletes’ time perception because it relies heavily on cognitive abilities, we selected the bisection task to focus specifically on perceptual differences in implied motion.

## 2. Materials and Methods

### 2.1. Participants

In this study, 20 college student athletes (including soccer, tennis, badminton, swimming, gymnastics, boxing, and wrestling) with a master or first grade (15 females; mean age = 19.4 ± 1.1 years old; sports training years = 8.1 ± 2.7) and 20 college students who are not athletes (18 females; mean age = 20 ± 1.8 years old) were recruited as participants from Beijing Sport University. No difference in age was detected between the athletes and non-athletes (*t* = 1.26, *p* = 0.219). All participants reported normal or corrected-to-normal visual acuity in both eyes. And they gave written informed consent to participate in the study in accordance with the procedure approved by the Ethics Committee of Beijing Sport University.

### 2.2. Experimental Stimuli

The stimuli were images of human characters with an implied motion (running or walking) or a static (standing) posture, displayed on a gray background (Figure 1), which were created using Easy Poser software (v1.5.53). Because we wanted to present daily movement rather than sports situations, the characters of the images were wearing school uniforms rather than tracksuits, and we did not recruit runners in this study. We created twelve pictures, including four sets of pictures with each set containing a running, a walking, and a standing posture. The four sets were a female left (the human character was toward the left) set, female right (the human character was toward the right) set, male left set, and male right set. When we finally analyzed the data, one of the most important independent variables was the posture type (running vs. walking vs. standing), namely the stimulus type. Thirty-two participants (mean age = 20.1 ± 1.04 years old) were recruited to rate the movement intensity (Likert rating ranging from 0 to 10) of all the characters online by questionnaires. The results showed that the intensities of movement were significantly different among the three kinds of postures [running: 7.17; walking: 4.36; standing: 2.77; *F* (2, 62) = 92.35; *p* < 0.001; η^p2^ = 0.75].

The presentation of stimuli and collection of data were computer-controlled. And the whole experimental procedure was generated by E-prime 2.0.

### 2.3. Procedure

We adopted the temporal bisection task to measure the perceived duration of the image presentation [14]. The whole experiment was divided into two blocks, a sub-second experimental block and supra-second experimental block. The order of the two blocks was counter-balanced in both athlete and non-athlete groups, and there was a rest between the two blocks. And each block consisted of three phases, a learning phase, a training phase, and a test phase.

In the sub-second experimental block, firstly, participants needed to learn two standard durations. That is, in the learning phases, participants viewed a green square object in the center of the screen for durations of 0.3 s or 0.9 s in sequence, and they were also told which one was short or long in each trial. Then, in the training phase, participants were asked to judge whether the duration (0.3 s, 0.4 s, 0.5 s, 0.6 s, 0.7 s, 0.8 s, 0.9 s) of the stimulus (the same square as in the learning phase) was more similar to the long or to the short standard duration presented in the learning phase before by pressing the “d” or “k” key. The response keys were counterbalanced across participants. And visual feedback was provided after each trial. In the test phase, the stimuli were the character pictures containing running, walking, and standing postures. Each test stimulus was presented with seven probe durations (0.3 s, 0.4 s, 0.5 s, 0.6 s, 0.7 s, 0.8 s, or 0.9 s) after a fixation display. The judging task was the same as in the training phase without any feedback.

The procedure of the supra-second experimental block was similar to that of the sub-second experimental block. However, the two standard durations were 1 s and 1.6 s. And the probe durations were 1 s, 1.1 s, 1.2 s, 1.3 s, 1.4 s, 1.5 s, and 1.6 s. There were 84 trials in the test phase, and the trial order was randomized across participants and across blocks.

### 2.4. Data Analysis

The proportions of “long” responses were calculated for each probe duration and each stimulus type. Cumulative Gaussian psychometric functions were fitted separately to the proportion of long responses for the three types of images using the psignifit toolbox for Matlab, which implements the maximum-likelihood method. The bisection point (BP) was then calculated based on the 50% point in the obtained logistic curve to compare the mean perceived duration of the test stimuli. To further examine the precision or sensitivity of time perception, we calculated the just-noticeable-difference (JND) of the temporal bisection using half of the difference in duration between the 25 and 75% points [15].

## 3. Results

### 3.1. Sub-Second Block

First, we conducted a three (stimulus type: running, walking, and standing) × two (participant type: athlete and non-athlete) repeated-measures ANOVA on the bisection point. The results showed a significant main effect of stimulus type [*F* (2, 76) = 8.67; *p* < 0.001; η^p2^ = 0.19]. Post hoc analysis showed that the BP for stimuli with implied motion was significantly different from that of the stimuli with static posture. To be more specific, the BP for a running posture (mean = 583.13 ms; SE = 67.84 ms) was significantly smaller (*p* = 0.003) than that of a standing posture (mean = 612.08 ms; SE = 57.46 ms) and was not significantly different (*p* = 0.68) from that of a walking posture (mean = 585.95 ms; SE = 56.91 ms). And the BP for a walking posture was also significantly smaller (*p* < 0.001) than that of a standing posture. However, the main effect of participant type was not significant [*F* (1, 38) = 0.48; *p* = 0.49; η^p2^ = 0.01]. In addition, the interaction of the stimulus type and participants type was not significant either [*F* (2, 76) = 0.13; *p* = 0.88; η^p2^ = 0.003]. The mean proportion of long responses and the mean bisection point are shown in Figure 2.

Next, we carried out the same repeated-measure ANOVA on the JND. However, neither the main effect of stimulus type [*F* (2, 76) = 2.37; *p* = 0.1; η^p2^ = 0.06] and participant type [*F* (1, 38) = 0.51; *p* = 0.48; η^p2^ = 0.01] nor of interaction effects [*F* (2, 76) = 2.38; *p* = 0.1; η^p2^ = 0.06] were significant.

### 3.2. Supra-Second Block

The data analysis procedure for the supra-second block was similar to that of the sub-second block. First, the results of the repeated-measures ANOVA on the BP revealed a significant main effect of stimulus type [*F* (2, 76) = 10.34; *p* < 0.01; η^p2^ = 0.21]. The post hoc analysis showed that the BP for stimuli with implied motion was significantly different to stimuli with static posture. To be more specific, the BP for a running posture (mean = 1288.4 ms; SE = 84.23 ms) was significantly smaller (*p* < 0.001) than that of a standing posture (mean = 1342.01 ms; SE = 93.9 ms) and was not significantly different (*p* = 0.19) from that of a walking posture (mean = 1300 ms; SE = 64.59 ms). And the BP for a walking posture was also significantly smaller (*p* = 0.003) than that of a standing posture. And the main effect of participant type was also significant [*F* (1, 38) = 5.81; *p* = 0.021; η^p2^ = 0.13]. The post hoc analysis showed that the BP for athletes (mean = 1286.7 ms; SE = 80.23 ms) was significantly smaller than that for non-athletes (mean = 1336.2 ms; SE = 75.13 ms). In addition, the interaction of stimulus type and participant type was not significant [*F* (2, 76) = 0.83; *p* = 0.44; η^p2^ = 0.02]. The mean proportion of long responses and the mean bisection point are shown in Figure 3.

Next, we carried out the same repeated-measure ANOVA on the JND. However, neither the main effect of stimulus type [*F* (2, 76) = 0.56; *p* = 0.58; η^p2^ = 0.01] and participant type [*F* (1, 38) = 1.56; *p* = 0.22; η^p2^ = 0.04] nor of interaction effects [*F* (2, 76) = 1.31; *p* = 0.28; η^p2^ = 0.03] were significant.

## 4. Discussion

The results indicated that the bisection functions of implied motion images for both sub- and supra-second durations significantly shifted toward the left, with a lower BP for the postures requiring greater movement compared to those requiring no movement. A lower BP means that participants tend to categorize a shorter duration as being “long”. This indicates that the perceived duration is longer than it objectively is, meaning that subjective time has expanded or been overestimated. Thus, in other words, subjective duration perception was more expanded for the more dynamic body postures than for the less dynamic ones. Additionally, athletes judged supra-second durations to be longer than non-athletes did.

### 4.1. Duration Dilation Effect of Implied Motion

Our findings corroborated the hypothesis that the perceived duration in both the sub- and supra-second range was overestimated when observing body postures involving a greater degree of movement. This aligns with findings from Nather and Bueno [14] and Nather et al. [16], which suggested that viewing images of rapid movement prompts a faster counting pace within the internal clock, leading to an overestimation of the passage of time [17].

This phenomenon can be explained within the framework of embodied cognition, which posits that cognitive processes are closely intertwined with sensorimotor experiences. Firstly, the effects of implied motion on perceived duration may be mediated by the activation of motor cortex neurons that are involved in the performance of the actions depicted in the images. There is growing evidence that observing another’s action activates the same neurons involved in execution of the action, which are called “mirror neurons” [18]. Previous studies found that the mirror system responds not only to the observation of real actions but also to the observation of static images that depict implied bodily actions [19,20]. Thus, when implied motion is perceived, the brain activates the same perceptual systems as it does during actual physical movement, resulting in a similar time dilation effect. Secondly, the process of viewing images that imply movement can shift our attention toward dynamic features. Castellotti et al. demonstrated that viewing implied motion triggers pupil dilation, which suggests a level of attention enhancement and perceptual engagement [21]. When attention is directed toward motion, it may create a perception of longer intervals due to the increased cognitive load associated with processing complex visual information.

In light of these observations, our study verified that images depicting large human movements were judged to last longer, because processing them involves the embodied simulation of more effortful movements. However, we have to acknowledge that this theory provides a broad framework that can accommodate diverse results. We therefore interpret our results as offering context-specific support for embodiment, rather than definitive evidence of its mechanisms across different implied motion contexts.

We did not find significant differences in the JND among the test stimuli. This result was consistent with the findings of Yamamoto and Miura [5], as well as Nather et al. [16], suggesting that the temporal sensitivity of implied motion images with greater movement intensity did not differ from that of images implying less motion.

### 4.2. Impact of Long-Term Sports Training on Duration Perception

Another important finding of this study was that when viewing supra-second implied motion, the dilation effect in athletes with long-term sports training was greater than that in non-athletes. This results partially aligns with the main findings from Jia, Zhang, and Feng [10], which suggested that divers reproduced longer durations for diving stimuli compared to general stimuli in both the sub-second and supra-second time ranges, while non-athletes exhibited the opposite result. Our study extended the dilation effect observed in athletes from expertise-related tasks to those involving daily movement duration perception. The time perception advantage that athletes exhibit in specific sports contexts may transfer to their daily lives, enabling them to evaluate time more effectively in everyday activities. This transfer can enhance their decision-making abilities and reaction times in non-competitive environments. And the effect may reveal that long-term athletic training can enhance temporal perception abilities, even for general, non-expertise-related stimuli. This suggests potential applications in developing training programs aimed at improving cognitive functions linked to time perception, benefiting both athletes and non-athletes in various real-world settings.

A potential explanation for this finding is that enhanced information extraction efficiency among sports experts stems from their cognitive advantages. Consequently, the perceptual, attentional, and memory-related benefits associated with athletic training facilitate more effective information extraction, leading athletes to perceive time intervals as longer than non-athletes do. Supporting this view, several studies have demonstrated that improved attention and memory are linked to prolonged duration perception [22].

Our study only found a larger dilation effect for athletes in the supra-second time range, which is consistent with our hypothesis. Previous research indicated that sub-second and supra-second duration perception involve distinct mechanisms: Sub-second time perception is typically considered to be driven by sensory or automatic processes, which rely on lower-level neural systems such as the cerebellum and basal ganglia. These processes are associated with motor control and rapid responses and generally do not involve a significant cognitive load [23]. In contrast, supra-second time perception engages higher-order cognitive processes, such as attentional allocation and working memory, which are regulated by the parietal and prefrontal cortices. These regions are closely linked to complex cognitive functions, including decision-making and information integration [12]. However, Jia, Zhang, and Feng found a larger dilation effect for athletes in both time ranges [10]. The primary difference in our study is that the stimuli were not expertise-related. We speculate that long-term sports training may influence daily movement duration perception more prominently in the supra-second time range, which relies more heavily on attention and working memory, and needs efficient information extraction capabilities in dynamic environments. The observed phenomenon in this study, where athletes showed greater time dilation effects even with non-sport-specific implied motion stimuli, can be attributed to the generalization of cognitive benefits from athletic training. Long-term training enhances attentional and information processing abilities that extend beyond sport-specific contexts. Additionally, according to the embodied cognition theory, the extensive physical training that athletes undergo enhances their sensitivity to bodily-related visual cues, even when those cues are not directly related to their specific sports. This suggests that the neural mechanisms involved in processing dynamic visual information, such as those in the parietal and prefrontal cortices, are engaged more robustly in athletes, leading to pronounced time perception effects even with non-specific implied motion stimuli.

Another interesting controversy is that some studies found that long-term sports training will enhance the temporal accuracy, while our results found a dilation effect. And the Bayesian cue combination framework can help explain these phenomena by emphasizing the dual role of movement in time perception—it can both enhance precision and induce temporal distortions [24]. The framework explains why movement can improve the accuracy of time perception in certain situations—the brain combines the low-variance time estimates derived from movement with other sensory information to form a more precise overall time perception. In contrast, when the reliability of the movement is reduced or the perceptual task requires non-synchronous information, movement can lead to a distortion in time perception. In summary, how movement affects time perception depends on the specific properties of the movement, the type of sensory input, and the relationship between the movement and these inputs. Therefore, athletes have more precise time perception, because they can effectively integrate motor feedback and sensory information. Meanwhile, when observing sports or movement information, their brains, due to the anticipation of movement, experience temporal dilation as a unique way of integrating time and movement information.

### 4.3. Limitations

The present study has three limitations. Firstly, we did not distinguish the athletes’ specialty. Athletes engaged in open-skill sports are more influenced by environmental changes in their time duration, and they tend to exhibit stronger temporal regulation abilities in rapidly changing scenarios. In contrast, athletes in closed-skill sports, where the environment is more stable, rely more on their internal rhythm for time perception [25]. For example, in Perrone et al.’s study, they found that closed-skill sports, specifically time-related disciplines, enhance motor imagery and time perception abilities [26]. Therefore, future research should further explore the impact of different athletic specializations on the time perception of implied motion to deepen the understanding of the relationship between sports and cognition.

Secondly, due to the complexity of recruiting a certain number of competitive athletes, athletes in the study partially differed in terms of years of expertise and training intensity. And we cannot exclude that these variables may have influenced the observed findings. Combined with the above limitation, future research should fully consider the individual differences in professional years of athletes when distinguishing different specialized athletes.

Thirdly, the lack of control stimuli (e.g., scrambled or rotated images) limits the isolation of implied motion effects from other factors such as posture geometry and visual orientation. The different orientations and postures (e.g., horizontal vs. oblique leg orientations) in our stimuli may have influenced the perceived motion intensity and, thus, the duration perception. Research by Lo et al. suggests that angular configurations in walking and running postures could contribute to a sense of dynamism, which might in turn affect time estimation [6]. We suggest that future studies consider these structural elements and their potential influence on implied motion perception. And future studies could employ control conditions inspired by Arnheim [27] and Turvey [28] to assess the impact of geometric stability, spatial orientation, and posture angle on perceived duration, which may help clarify whether dynamic properties or visual orientation plays a primary role. For example, we can cut up the pictures so that features are present but in disarray, presented upside-down, or turned 90 degrees. And we can also add shadows underfoot and a track of implied motion to strengthen the stability of postures.

## 5. Conclusions

This study verified that the duration was judged to be longer for the more dynamic body posture than for the static one. The study also demonstrated that athletes exhibit a significantly greater dilation effect in duration perception for images depicting implied motion compared to non-athletes in the supra-second time range. Furthermore, the results indicate that duration perception in the supra-second range is more reliant on attention and working memory, aligning with athletes’ superior information processing capabilities in dynamic contexts. Long-term athletic training can enhance these cognitive advantages of duration perception, even in daily movement processing.

## Figures and Tables

**Figure 1 behavsci-14-01092-f001:**
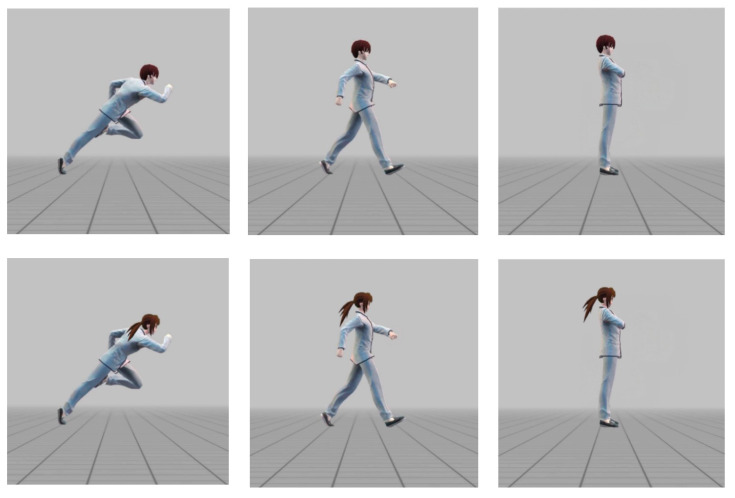
Illustration of the postures including running, walking, and standing. We only presented examples moving in the right direction.

**Figure 2 behavsci-14-01092-f002:**
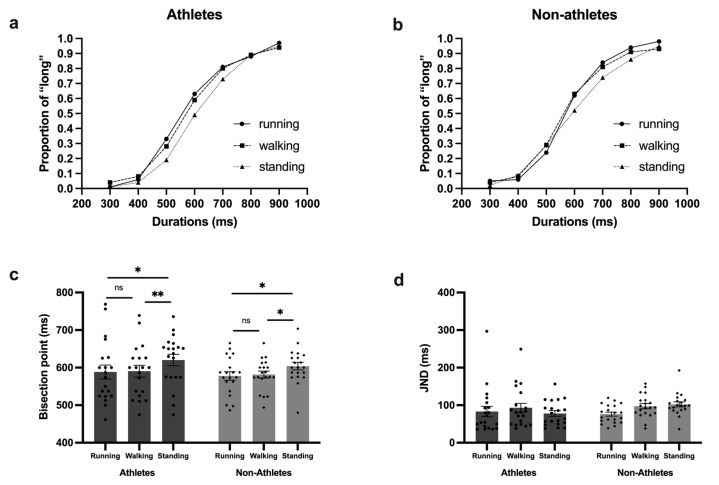
Results for the sub-second block. (**a**) The mean proportion of long responses of athletes; (**b**) the mean proportion of long responses of non-athletes; (**c**) the mean bisection points of athletes and non-athletes; (**d**) the mean JND of athletes and non-athletes. * *p* < 0.05; ** *p* < 0.01; ns: not significant.

**Figure 3 behavsci-14-01092-f003:**
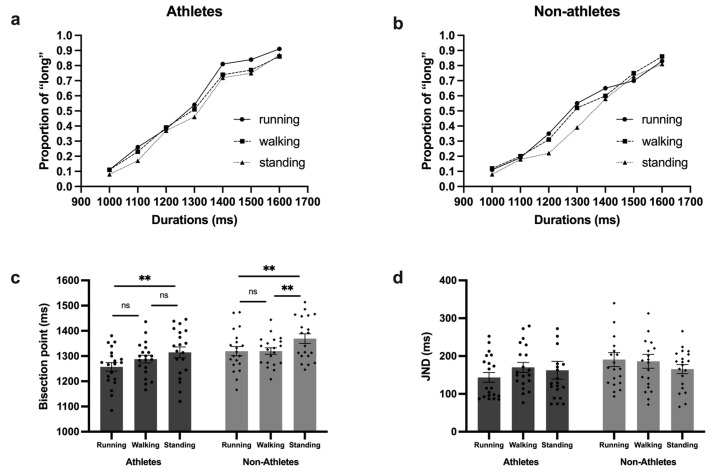
Results for the supra-second block. (**a**) The mean proportion of long responses of athletes; (**b**) the mean proportion of long responses of non-athletes; (**c**) the mean bisection points of athletes and non-athletes; (**d**) the mean JND of athletes and non-athletes. ** *p* < 0.01; ns: not significant.

## Data Availability

The data presented in this study are available on request from the corresponding author due to privacy.

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
