# Peer review of "Sub- and Supra-Second Duration Perception of Implied Motion: Differences Between Athletes and Non-Athletes"

_behavsci, 2024, doi:10.3390/bs14111092_

Round 1

Reviewer 1 Report

Comments and Suggestions for Authors

In the present paper the author investigated the differences between athletes and non-athletes in duration perception of implied motion. In particular the experimental design distinguished between sub-second and supra-second time perception by varying the duration of the stimuli in two separate blocks. The results showed that implied motion led to and over estimation of perceived duration in both the sub-second and supra-second range. The results also show a difference in the duration perception between athletes and non-athletes; however, this is only true in the supra-second range.

While I find the experimental design and analyses conducted to be straightforward and well executed, I have some minor comments that need to be addressed before the paper is ready for publication, especially regarding the hypothesis and the interpretation of the results. I also find the introduction and discussion to be a bit hurried, and they would benefit from a deeper discussion of previous literature.

1. In the Introduction the author states that athletes’ duration perception is characterized by higher accuracy and lower variability. How does this statement reconcile with the findings in the paper? If athletes are more likely to overestimate implied motion duration perception this means that they show lower accuracy compared to non-athletes, while based on this sentence in the Introduction we are led to expect the opposite.

2. Why does the author hypothesize that duration perception in athletes should be enhanced only in the supra-second range? It is not clear why sports training should impact only cognitive time perception processes and not automatic time perception. A prediction in this direction should be adequately justified by the literature.

3. Furthermore, the author states that a time bisection task was chosen because it less prone to cognitive distortions compared to a time reproduction task. However, if athletes performance in a duration perception task is influenced by cognitive processes (see point 3), shouldn’t a task that is more prone to this kind of distortions be used?

4. Figure 2C. I suggest adding a scatter of individual bisection points along with the average bars. On top of this, statistically significant differences between the bars should be depicted with asterisks, to aid with an easier interpretation of the results.

5. Even though no statistical difference in the analyses of the JND was found, a plot showing average performance across groups and conditions would still be very informative for the reader.

6. At line 198 and following the author cites Castellotti et al. (2021) to support the claim that “When implied motion is perceived, the brain ac-198 tivates the same perceptual systems as it does during actual physical movement, resulting 199 in a similar time dilation effect”. However, in my understanding the cited paper only demonstrates that there is a pupil dilation effect when implied motion is perceived, and the hypothesis that the same perceptual mechanisms are activated is a purely speculative interpretation of the results. I suggest the author revise their take on this paper to only include the interpretation of the results that is supported by the data presented in Castellotti et al. (2021). Furthermore, the authors did not investigate whether the pupil dilation induced by implied motion led to an overestimation of perceived duration. For this reason, also the claim at line 200 should be revised.

7. At line 202 the author claims that the perception of implied motion images is associated with greater arousal, and this leads to duration overestimation. Again, I find this interpretation to be too speculative in light of the results presented. While it is true that an overestimation of perceived duration was observed, the author did not implement any measurement of arousal, either directly or indirectly, meaning that it is not possible to conclude that the participants were more aroused when perceiving implied motion stimuli. On top of this, since athletes tend to over estimate duration more, this would imply that they are more aroused by implied motion perception, which sounds counterintuitive as sport training should render motion less effortful for athletes.

8. In general, I find the discussions and the bibliography cited to be a little hurried. The discussion about whether sub-second and supra-second intervals perception is performed by the same perceptual mechanisms of by separate pathways is vast, along with the discussion about which factors influence which interval. I suggest the author to complete the paper with a deeper discussion of this topic, starting for instance with Rammsayer 1991.

Bibliography:
Rammsayer T. H., Lima S. D. (1991). Duration discrimination of filled and empty auditory intervals: cognitive and perceptual factors.
Percept. Psychophys. 50 565–574. 10.3758/BF03207541

Reviewer 2 Report

Comments and Suggestions for Authors

Please consider alternative explanations and possible control stimuli with dynamic properties for perception, as R Arnheim has pointed out.

In this paper, observers who are athletes or not athletes are shown pictures of people standing, walking and running. Some pictures were shown very briefly (less than a second) and some slightly longer (more than a second). The brief-exposure walking and running pictures are judged by both athletes and non-athletes, equally, to be observed for longer periods than the standing pictures. The more-than-a second pictures were judged similarly, but in this case the athletes required less time for pictures (mean = 1286.7 ms, SE = 164 80.23 ms) than non-athletes (mean = 1336.2 ms, SE = 75.13 ms). In addition, the interaction of stimulus type and participants type was not significant [F 166 (2, 76) = 0.83, p = 0.44, ηp2 = 0.02].

In sum: Motion-implying pictures, shown briefly, seem to have a longer duration than ones implying a static pose.  Further, if the pictures are shown for slightly longer (over a second) they seem longer to athletes than non-athletes. 

The ms has enough conditions  -- short & long duration, 3 poses, athletes and non-athletes – to merit publication.  The effects are small, but significant. The reason for the effects is not wel—established.

In the Discussion, the results are said to favour embodiment theory, i.e. that our perception is intertwined with our body activities. However, the theory makes no specific prediction. Any result with a significant difference would be said to favour embodiment theory, It is too general to make a specific prediction. Indeed, if the differences between stimuli were in the opposite direction the same general claim would be made. This section of Discussion cannot remain as is.

We do not know that motion is actually involved as there is little in the way of controls.  The 3 postures are different geometrically e.g. the running pose is oblique, the walking pose has V shapes formed by the legs, only part of one arm and one vertical leg is present in the standing pose. The ms should point out that the relevant feature affecting the result is open for discussion debate. There are no control pictures. There is no discussion of ways that control pictures might be designed e.g. cutting up the pictures so features are present but in disarray, presented upside-down, or turned 90 degrees. The standing person could have arms posed so as to be visible. The running person could be less oblique. The running person is in an unstable pose, not controlled-for. 

No shadows are present underfoot. The figures may appear to some observers to be floating. There is no track on which the characters are active, just an empty plane. The energy required for the characters and their actions involves pushing off from the ground in the walking and running poses. The standing pose has energy flowing vertically if there is a ground, but in a stable manner. I recommend reading M. Turvey on dynamics and perception. 

The feet of the characters are in very different orientations -- horizontal, oblique and vertical. Are these orientations a key factor? Would horizontal lines seem stable, and obliques to be dynamic? Is this a key factor? Are near-verticals seen as unstable, toppling? Only walking reveals two feet. There are more angles in the walking and running poses. Are angular pictures see as dynamic? I recommend reading R. Arnheim.

The ms is well written – very smooth, excellent English -- and it cites relevant research on motion-implying pictures. Its data analysis is acceptable. It is light on controls and leaps to conclusions without considering alternative accounts.
